# GROUP CRITICAL-TOKEN POLICY OPTIMIZATION FOR AUTOREGRESSIVE IMAGE GENERATION

**Guohui Zhang[1], Hu Yu[1], Xiaoxiao Ma[1], Jinghao Zhang[1,2], Yaning Pan[3], Mingde Yao[4], Jie Xiao[1], Linjiang Huang[5], Jie Huang[1], Feng Zhao[1†]**
[1]University of Science and Technology of China, [2]Shanghai Innovation Institute
[3]Fudan University, [4]The Chinese University of Hong Kong, [5]Beihang University

## ABSTRACT

Recent studies have extended Reinforcement Learning with Verifiable Rewards (RLVR) to autoregressive (AR) visual generation and achieved promising progress. However, existing methods typically apply uniform optimization across all image tokens, while the varying contributions of different image tokens for RLVR's training remain unexplored. In fact, the key obstacle lies in how to identify more critical image tokens during AR generation and implement effective token-wise optimization for them. To tackle this challenge, we propose **G**roup **C**ritical-token **P**olicy **O**ptimization (**GCPO**), which facilitates effective policy optimization on critical tokens. We identify the critical tokens in RLVR-based AR generation from three perspectives, specifically: **(1)** Causal dependency: early tokens fundamentally determine the later tokens and final image effect due to unidirectional dependency; **(2)** Entropy-induced spatial structure: tokens with high entropy gradients correspond to image structure and bridges distinct visual regions; **(3)** RLVR-focused token diversity: tokens with low visual similarity across a group of sampled images contribute to richer token-level diversity. For these identified critical tokens, we further introduce a dynamic token-wise advantage weight to encourage exploration, based on confidence divergence between the policy model and reference model. By leveraging 30% of the image tokens, GCPO achieves better performance than GRPO with full tokens. Extensive experiments on multiple text-to-image benchmarks for both AR models and unified multimodal models demonstrate the effectiveness of GCPO for AR visual generation. Code is available at https://github.com/zghhui/GCPO

## 1 INTRODUCTION

Visual generative models (Sun et al., 2024; Liu et al., 2024a; Ma et al., 2024) based on the autoregressive (AR) paradigm have made significant progress in the field of high-quality image generation. Meanwhile, Reinforcement Learning (RL) with Verifiable Rewards (RLVR), demonstrated by OpenAI-o1 (OpenAI, 2024) and DeepSeek R1 (Guo et al., 2025a) to enhance the reasoning abilities of large language models (LLM) (Yang et al., 2025; Team et al., 2025), is now being gradually introduced into the visual generation to improve preference alignment and task controllability.

Recent works apply RLVR, especially Group Relative Policy Optimization (GRPO) (Shao et al., 2024) for text-to-image generation by designing visual Chains-of-Thought (CoT) (Jiang et al., 2025), optimizing reward functions (Yuan et al., 2025), and constructing customized datasets (Pan et al., 2025a). Despite these advances, these methods typically assume that each token contributes equally to the RLVR's training objective and apply uniform policy optimization across the entire image token sequence. While different tokens play distinct roles in text-to-image generation: some tokens determine and correspond to the global structure of the image, while others correspond to backgrounds or details. Concurrent RLVR-based LLM reasoning works (Wang et al., 2025d;a) also realized such a functional distinction between tokens, and split them into reasoning-related *critical tokens* and remaining knowledge-related tokens, where the former have higher entropy and dominate reasoning

---

[†]Corresponding author.

ability. While this analogy highlights a shared imbalance in token importance, visual generation bears higher complexity due to the causal AR modeling and bidirectional image structure.

In this paper, we identify **critical tokens** in RLVR-based AR generation from three perspectives: 1) **Causal dependency** of AR; 2) **Entropy-induced spatial structure**; and 3) **RLVR-focused token diversity**. Specifically, **1)** Early generated tokens continuously influence subsequent tokens and have a significant impact on the overall image structure due to the causal attention mechanism, as shown in Fig. 1. **2)** For the token sequence of each image, we initially attempt to correlate entropy with image token, similar to the operations in (Wang et al., 2025d). While, we find that the distribution of high/low entropy tokens didn't consistently correspond to certain parts of the image, like structure or background. Pushing one step further, we observe that the token entropy gradient map demonstrates a consistent spatial pattern among images, with the high value approximately corresponding to the structures and bridging distinct visual regions, see Fig. 2, which is sensitive to RL. **3)** Within the group of GRPO, we observe that tokens deliver varying diversities for the same position along images. As shown in Fig. 3, tokens corresponding to background and texture regions tend to exhibit higher similarity, while tokens with lower similarity correspond to more complex region structures.

On the basis of our critical token selection strategy, we introduce **G**roup **C**ritical-token **P**olicy **O**ptimization (**GCPO**), a novel RLVR framework for AR image generation that facilitates effective policy optimization on critical image tokens. During each optimization step, GCPO first selects the critical tokens following the above three perspectives. Then, we further devise a dynamic token-level advantage weight for critical tokens to better encourage exploration. This dynamic weight is based on confidence divergence of critical tokens between the updating policy model and reference model, which differs from the standard GRPO that allocates advantage uniformly for each token. Finally, we only retain the policy gradients of critical tokens to perform policy optimization.

By utilizing only critical tokens (**30%** of the tokens), GCPO achieves better performance than GRPO with full tokens on multiple text-to-image generation benchmarks. Extensive experiments demonstrate the effectiveness of GCPO, including Geneval, T2I-CompBench, and Human Preference Benchmark, which is also verified on both AR models and unified multimodal models. In summary, our contributions are as follows:

- We identify **critical tokens** in RLVR-based AR visual generation from three perspectives: Causal dependency of AR, Entropy-induced spatial structure, and RLVR-focused token diversity, achieving a structure-centric and comprehensive critical-token selection strategy.

- We propose **Group Critical-token Policy Optimization (GCPO)**, a RLVR framework for autoregressive image generation that facilitates effective policy optimization on critical image tokens. We further devise the dynamic advantage weight strategy for critical tokens to enable reasonable exploration and stabilize the inherently generative prior, based on their confidence divergence between the policy model and the reference model.

- GCPO is applicable to both AR models and unified multimodal models. Extensive experiments demonstrate that GCPO, by optimizing only critical tokens (**30%** of the tokens), achieves better performance than GRPO with full tokens across multiple T2I benchmarks.

## 2 RELATED WORK

**Autoregressive Visual Generation**. Autoregressive image generation models (Sun et al., 2024; Team, 2024; Ma et al., 2024; Pan et al., 2025b) adopt the next token prediction paradigm, which has been widely applied in LLMs, to enable text-to-image generation. Representative works (Ramesh et al., 2022; Liu et al., 2024a) typically first use an image tokenizer (Esser et al., 2021) to discretize continuous image data into a sequence of tokens, and then employ a transformer architecture to model visual tokens. Furthermore, recent research (Huang et al., 2025) has focused on unifying image generation and image understanding within a single architecture. These models are capable of accepting diverse types of input (e.g., text, image, video) and producing one or more modalities as output. MetaMorph (Tong et al., 2024) utilizes SigLIP to extract visual embeddings and introduces modality-specific adapters for more efficient cross-modal alignment. Janus-Pro (Chen et al., 2025a) adopts a dual-encoder structure that separately processes textual and visual data. Transfusion (Zhou et al., 2024), Show-o (Xie et al., 2024), and BAGEL (Deng et al., 2025) further combine the strengths

of Transformer and Diffusion architectures for multimodal understanding and generation, achieving superior performance across various tasks.

**Reinforcement Learning for Visual Generation**. Reinforcement Learning with Verifiable Rewards (RLVR) has achieved significant progress in the field of large language models (LLMs). A series of open-source models (Guo et al., 2025a; Team et al., 2025) and RLVR methods (Yu et al., 2025c; Yue et al., 2025; Shrivastava et al., 2025; Zeng et al., 2025a; Pan et al., 2025c) have been proposed, further advancing the development of this field. Meanwhile, recent efforts (Liu et al., 2025; Xue et al., 2025; Pan et al., 2025a) have increasingly explored the potential of RLVR, especially Group Relative Policy Optimization (Shao et al., 2024) (GRPO), in the field of visual generation. SimpleAR (Wang et al., 2025b) has demonstrated that GRPO can significantly enhance the aesthetic quality and prompt alignment of AR models. T2i-R1 (Jiang et al., 2025) leverages GRPO to jointly optimize both semantic-level and token-level Chain-of-Thought (CoT) reasoning processes, thereby improving the generative capabilities of a unified multimodal model. In addition, (Gallici & Borde, 2025) introduces GRPO into scale-wise visual generative models that achieve high-quality image generation by predicting the "next scale" and demonstrating great potential (Han et al., 2025).

**Reinforcement Learning for Critical tokens**. Recently, several studies (Li et al., 2025; Vassoyan et al., 2025; Wang et al., 2025c; Zhao et al., 2025; Zeng et al., 2025b; Ma et al., 2025) have focused on the deeper analysis and exploration of the role of RL in LLMs' reasoning task, especially at the token level. Critical Tokens Matter (Lin et al., 2024) suggests that identifying and replacing critical tokens can significantly improve the model's accuracy, and proposes a contrastive estimation method to accurately locate these tokens. ConfPO (Yoon et al., 2025) further investigates the effectiveness of selectively optimizing only low-confidence and information-rich tokens. Additionally, (Wang et al., 2025d;a) points out the existence of "fork" tokens in LLM reasoning paths, indicating that these tokens typically have high entropy and are related to logical reasoning. They further observe that these fork tokens (critical tokens) can be identified by entropy and play more critical role than other tokens in enhancing the LLM's reasoning ability. However, identifying critical tokens in AR visual generation and implementing effective token-wise optimization for them remains unexplored.

## 3  PRELIMINARY

**Autoregressive Image Generation.** A common autoregressive (AR) model includes two main components: an image tokenizer and an AR transformer (Esser et al., 2021; Yu et al., 2021; 2025b;a). For image tokenizer, typically VQ-VAE (Van Den Oord et al., 2017), it converts images $\mathcal{I} \in \mathcal{R}^{H \times W \times 3}$ into discrete tokens sequence $Z = \{z_1, \ldots, z_N\}$, where each token $z_t \in \mathcal{V}$. $\mathcal{V}$ and $t$ represent the VQ-VAE codebook and token index in sequence, respectively. Next, the AR model autoregressively predicts the joint distribution of the next image token conditioned on the text and previously generated tokens: $P(z|c) = \prod_{t=1}^{N} P(z_t|z_{<t}, c)$, where $c$ represents text embedding.

**Token Entropy.** The AR model outputs probability distribution over the codebook $\mathcal{V}$ for each token. Therefore, we can use the following formula to calculate the entropy (Shannon, 1948) of this distribution, which is referred to as the token entropy $H$:

$$H(z_t) = -\sum_{k=1}^{V} P_{t,k} \log P_{t,k}. \tag{1}$$

**Group Relative Policy Optimization (GRPO).** For each prompt $p$, GRPO (Shao et al., 2024) samples a group of image outputs $\{o^1, o^2, \ldots\}$ from the old policy model $\pi_{\theta_{old}}$. Then, it computes the corresponding rewards $\{r^1, r^2, \ldots\}$ for each output within each group. The advantage $A^i$ is calculated from the rewards of the group, and each output in the group shares the same advantage. Then optimizes the policy model $\pi_{\theta_{old}}$ by maximizing the following objective:

$$\mathcal{J}_{\text{GRPO}}(\theta) = \mathbb{E}_{\{o^i\}_{i=1}^{G} \sim \pi_{\theta_{\text{old}}}}$$

$$\left[ \frac{1}{G} \sum_{i=1}^{G} \frac{1}{|o^i|} \sum_{t=1}^{|o^i|} \left( \min\left( r_t^i(\theta) \hat{A}^i, \text{clip}\left( r_t^i(\theta), 1 - \varepsilon, 1 + \varepsilon \right) \hat{A}^i \right) - \beta D_{\text{KL}}(\pi_\theta \| \pi_{\text{ref}}) \right) \right], \tag{2}$$

where

$$A^i = \frac{r^i - mean(\{r^1, r^2, \cdots, r^G\})}{std(\{r^1, r^2, \cdots, r^G\})}, \quad r_t^i(\theta) = \frac{\pi_\theta(o_t^i \mid q, o_{<t}^i)}{\pi_{\theta_{old}}(o_t^i \mid q, o_{<t}^i)}. \tag{3}$$

## 4 OBSERVATION AND ANALYSIS IN AR VISUAL GENERATION

### 4.1 CAUSAL DEPENDENCY OF AR

The core capability of AR models stems from the causal attention mechanism in transformers. Under this next token prediction paradigm, the initially generated tokens continuously influence the generation of all subsequent tokens, thereby significantly impacting the overall structure and layout of the image. To further validate this point, we inject additional noise into the tokens at different positions during the generation process (Beyer et al., 2025). This causal influence is visibly illustrated in Fig. 1, where perturbations to the early 58 tokens (token index from 0 to 58) introduce substantial changes to the image's global structure, while perturbations to the middle 58 tokens (token index from 250 to 308) only affect local details. This empirical evidence confirms that early tokens serve as global priors and structural guides. In contrast, later tokens are constrained by both preceding tokens and local consistency, making them more focused on generating local content and details. There-

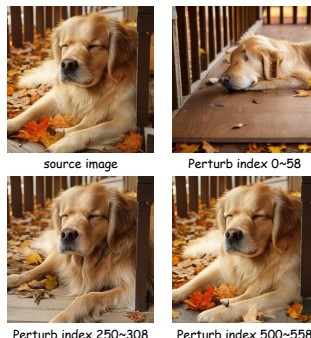

Figure 1: Visual results of perturbing different tokens.

fore, the initial tokens should be part of critical tokens, as these early decisions propagate throughout the entire AR generation and establish the foundation for high-quality visual structures.

### 4.2 ENTROPY-INDUCED SPATIAL STRUCTURE

Concurrent RLVR-based LLM reasoning works (Wang et al., 2025d;a) have found a functional distinction between text tokens, and split them into reasoning-related critical tokens and knowledge-related tokens based on *token entropy*. The former with high entropy (such as "wait", "however") serve as logical **connectors** that bridge consecutive reasoning parts in CoT, while the latter with low entropy primarily capture factual or domain-specific knowledge.

Inspired by this, we first analyze the entropy distribution of image tokens. As shown in Fig. 2, we observe that the distribution of high/low entropy tokens corresponds to different parts of the image in different prompts: tokens with high entropy mainly correspond to the background in Fig. 2 (a), while corresponding to the subject in Fig. 2 (b). This phenomenon gradually becomes obvious in RL training. We argue that it is likely influenced by image and prompt complexity, and we provide more analysis results in Sec. B. Furthermore, we find that this entropy distribution in images exhibits a regional and spatial pattern, where tokens within the subject or background regions have approximate entropy values. We argue that image tokens exhibit strong spatial locality, with neighboring pixels sharing similar visual characteristics and entropy values (He et al., 2024; Xiang & Fan, 2025).

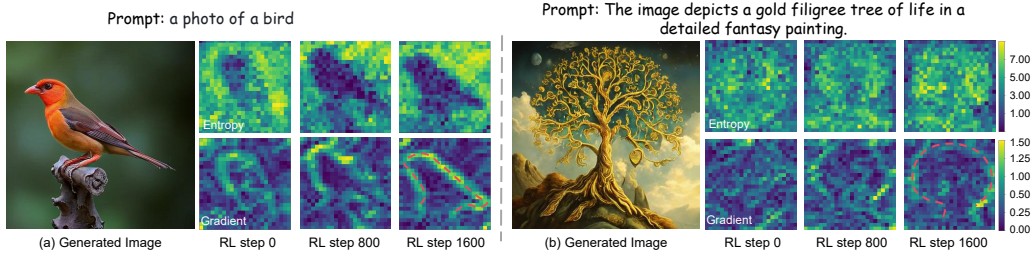

Figure 2: The entropy maps of images exhibit distinct spatial patterns. Tokens with high entropy gradient consistently correspond to image structure, which gradually strengthens with RL training.

Pushing one step further, we analyze the 2D gradient of entropy and observe consistent spatial patterns among different images. Specifically, tokens with high entropy gradients typically correspond to subject structure or regions with significant structural variation, and this pattern becomes more pronounced with RL training. These tokens exhibit large entropy changes in neighboring tokens, serving as **connectors** to link distinct visual regions. Based on this insight, we point out that tokens with high entropy gradients are critical and sensitive for spatial structures, and entropy gradients can serve as a universal and reliable proxy for identifying tokens associated with image structure.

### 4.3 RLVR-FOCUSED TOKEN DIVERSITY

GRPO typically relies on sample-level reward signals, where the differences between samples guide the direction of policy optimization. A group of similar samples provides limited reward information, which restricts the model's performance improvement and training efficiency. In light of this, (He et al., 2025; Yu et al., 2025c; Chen et al., 2025b) focus on enhancing sample diversity through entropy regularization or encouraging semantic diversity. In AR visual generation task, we further focus on token-level diversity. Within the group of GRPO, we observe that tokens deliver varying diversities for the same position along images. As shown in Fig. 3, tokens in background and texture regions tend to exhibit higher

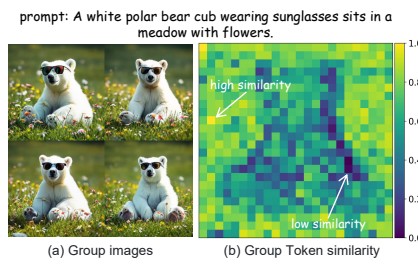

Figure 3: The average cosine similarity of tokens at corresponding location across a group of images.

similarity and hardly reflect the visual differences among images. While tokens with lower similarity correspond to more complex regional structures and contribute to richer information for GRPO-based policy optimization. Therefore, we select tokens with low similarity as part of critical tokens.

### 4.4 DYNAMIC ADVANTAGE WEIGHT

Balancing exploration with stabilization of generative priors is crucial for different tokens during RLVR training (Wang et al., 2025a). We argue that different critical tokens should have dynamic and distinct exploration constraints. Specifically, initial tokens should explore more moderately to prevent global structural collapse, while tokens with high entropy gradients and low similarity should have stronger exploration. Considering that the reference model itself serves as the starting point for training, we analyze the confidence divergence of each token between the training policy model and the reference

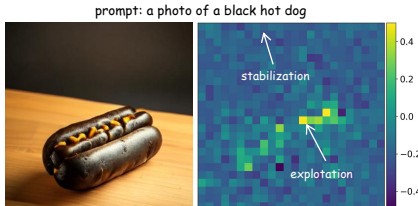

Figure 4: Confidence divergence between the Policy and Reference model

model. As shown in Fig. 4, we observe that this divergence is not only dynamic as the policy model updating, but also has distinct constraints: initial tokens exhibit smaller confidence divergence, while tokens corresponding to structures show larger divergence. Based on this, we utilize this dynamic divergence as weight to eliminate complex manual specification.

## 5 GROUP CRITICAL-TOKEN POLICY OPTIMIZATION

### 5.1 TOKEN SELECTION

The Sec. 4 discusses the critical image tokens selection from three aspects. Here, we detail the token selection strategy, as illustrated in Fig. 5. Specifically, we denote the image token sequence as $Z = \{z_1, \ldots, z_N\}$, where $N$ is the total length of the sequence. The initial image tokens is represented as $Z_{init} = \{z_1, \ldots, z_{K_{init}}\}$, with $K_{init}$ indicating the number of initial tokens.

Subsequently, we reshape the entropy sequence $\{H_t\}_{t=1}^N$ associated with these tokens into a 2D entropy map to select structure-related tokens. To mitigate the influence of noise in the entropy map, we perform a local averaging operation. Considering the local spatial and causal dependency of image tokens, the averaging is performed as follows:

$$\bar{H}_t = mean(H_t + H_t^{(l,u)} + H_t^{(u)} + H_t^{(r,u)} + H_t^{(l)}), \tag{4}$$

where $(l, u), (u), (r, u)$ and $(l)$ denote the upper-left, upper, upper-right and left neighboring positions of $H_t$, respectively. Then, we use the central difference to calculate the gradient of each token in the average entropy map. We select the $K_{struct}$ tokens with the largest gradients as $Z_{struct}$.

Next, we calculate the cosine similarity of token embeddings at each sequence position within group of images. Specifically, for each token sequence position $t$, we consider the group of token embeddings $\{e_{t,1}, e_{t,2}, \ldots, e_{t,G}\}$ derived from $G$ images. The pairwise cosine similarity between token

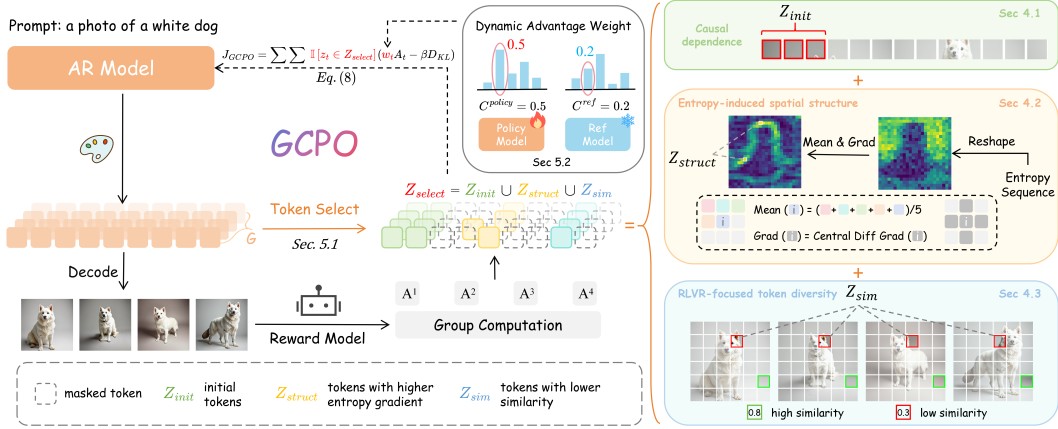

Figure 5: **Overview of GCPO.** GCPO first generates a group of images for each prompt and obtains the corresponding reward. For token selection, we first select the initial token as $Z_{init}$. Then, we calculate the local mean and central difference gradient in 2D entropy map, and select the token with the higher gradients as $Z_{struct}$. Subsequently, we select the tokens with lower similarity as $Z_{smi}$ based on intra-group cosine similarity at each position. For dynamic advantage weight, we calculate the cumulative mean confidence difference of each token as the advantage weight $w_t$. Finally, we only retain the policy gradients of critical tokens $Z_{select}$ to perform policy optimization.

embeddings at position $t$ is calculated as follows:

$$S_{jk}^{(t)} = \frac{e_{t,j} \cdot e_{t,k}}{\|e_{t,j}\|\|e_{t,k}\|}, \quad 1 \le j < k \le G. \tag{5}$$

We then calculate the average pairwise similarity $\bar{S}_t$ for each sequence position $t$. We select the $K_{\text{similarity}}$ tokens with the lowest average based on $\bar{S}$.

Finally, the overall critical token selection set is defined as the union of the three subsets:

$$Z_{select} = Z_{init} \cup Z_{struct} \cup Z_{sim}. \tag{6}$$

By default, the size of each subset $K_{init}$, $K_{struct}$, and $K_{sim}$ is set to 10% of the total token sequence length, ensuring balanced and representative selection.

## 5.2 DYNAMIC ADVANTAGE WEIGHT

The Sec. 4.4 discusses the motivation of the dynamic advantage weight. Furthermore, considering that the token at position $t$ is predicted by its preceding tokens, we further employ the cumulative average of confidence divergence as the weight for each critical token. This can be formalized as:

$$w_t = \frac{1}{t} \sum_{j=1}^{t} clip\left(C_j^{policy} - C_j^{ref}, -\epsilon_w, \epsilon_w\right), \tag{7}$$

where $w_i$ denotes the advantage weight at position $t$. $C_j^{policy}$ and $C_j^{ref}$ represent the confidence (model's log probability) of the $j$-th token on the policy model and the reference model, respectively. $\epsilon_w$ is the clip coefficient to prevent excessive weights from influencing training stability.

## 5.3 OBJECTIVE FUNCTION

The overall objective of GCPO is formulated as follows:

$$\mathcal{J}_{\text{GRPO}}(\theta) = \mathbb{E}_{\{o^i\}_{i=1}^{G} \sim \pi_{\theta_{old}}} \left[ \frac{1}{G} \sum_{i=1}^{G} \frac{1}{|o^i|} \sum_{t=1}^{|o^i|} \mathbb{I}\left[z_t \in Z_{select}\right] \left( w_t^i \min\left(r_t^i(\theta)\hat{A}^i, \right. \right.$$
$$\left. \left. \text{clip}\left(r_t^i(\theta), 1-\varepsilon, 1+\varepsilon\right)\hat{A}^i\right) - \beta \mathbb{D}_{\text{KL}}(\pi_\theta \| \pi_{\text{ref}}) \right) \right], \tag{8}$$

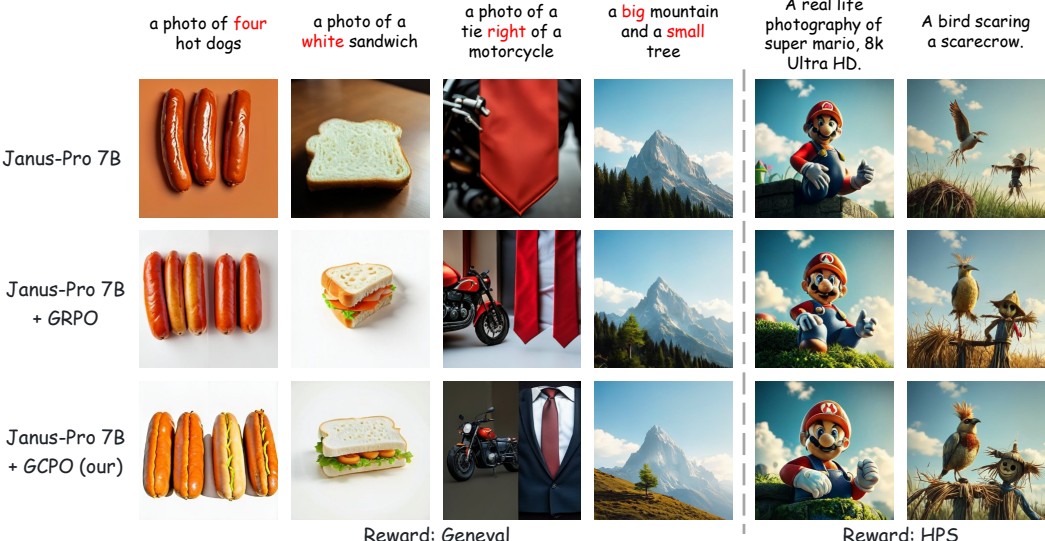

Figure 6: Visualization Results. We provide image generation results on Counting, Color, Position, and Shape tasks, as well as image quality.

where $\mathbb{I}[\cdot]$ is the indicator function that evaluates to 1 if the condition inside holds and 0 otherwise.

The differences are highlighted in red between Equation ( 8) and Equation ( 3): **(i)** The optimization term of each token is multiplied by $\mathbb{I}[z_t \in Z_{select}]$, ensuring that only critical tokens in $Z_{select}$ are involved in the overall optimization objective; **(ii)** The advantage term of each critical token is multiplied by $w_t$, allocating token-wise advantage weight to encourage exploration.

## 6 EXPERIMENTS

In this section, we evaluate GCPO to improve the performance of AR model and the unified multi-modal model on two representative tasks. (1) Composition Image Generation: We report the results on GenEval and T2i-Compbench, which primarily evaluate the models' ability on spatial relationships and color attributes. (2) Image quality and Human Preference Alignment: We report DEQA, ImageReward, and PickScore metrics, reflecting the visual quality and human preference of images.

### 6.1 EXPERIMENTAL SETUP

**Training Settings.** For the composition image generation task, our training dataset is sourced from 50,000 training prompts generated by Geneval pipeline, following (Liu et al., 2025). For the image quality and human preference alignment task, we utilize 15000 prompts from the HPSv2 training dataset. We conduct evaluations on LlamaGen (Sun et al., 2024), Janus-Pro 1B (Chen et al., 2025a), and Janus-Pro 7B (Chen et al., 2025a). Please refer to the Appendix A for detailed training settings.

**Benchmark.** We evaluate our method on Geneval (Ghosh et al., 2023), T2I-CompBench (Huang et al., 2023), and DrawBench (Saharia et al., 2022) to comprehensively validate effectiveness. T2I-CompBench is a comprehensive benchmark for open-world compositional text-to-image generation, covering six compositional task categories. DrawBench contains comprehensive and challenging prompts designed to assess the generative capabilities of T2I models. We use it to evaluate DEQA-Score (You et al., 2025), ImageReward (Xu et al., 2023) and Pick Score (Kirstain et al., 2023) metrics. We also report HPSv2 score (Wu et al., 2023) on HPSv2 Benchmark test data.

**Reward Model.** For the composition image generation task, following (Liu et al., 2025), we adopt Geneval reward as our reward model. For the image quality and human preference alignment task, we use HPSv2 as our reward model. Notably, given the relatively lower performance of LlamaGen on Geneval, we only use HPSv2 as the reward model for this base.

Table 1: Quantitative comparison results on the GenEval benchmark. The best result is in green .

| Method | Overall↑ | Sing Obj.↑ | Two Obj.↑ | Counting↑ | Color↑ | Position↑ | Color Attr.↑ |
|---|---|---|---|---|---|---|---|
| *Diffusion-based Method* | | | | | | | |
| PixArt-$\alpha$ (Chen et al., 2024) | 0.48 | 0.98 | 0.50 | 0.44 | 0.80 | 0.08 | 0.07 |
| SDXL (Podell et al., 2023) | 0.55 | 0.98 | 0.74 | 0.39 | 0.85 | 0.15 | 0.23 |
| SD3 (Esser et al., 2024) | 0.63 | 0.98 | 0.78 | 0.50 | 0.81 | 0.24 | 0.52 |
| DALL-E 3 (Betker et al., 2023) | 0.67 | 0.96 | 0.87 | 0.47 | 0.83 | 0.43 | 0.45 |
| FLUX.1-dev (Labs, 2024) | 0.66 | 0.98 | 0.81 | 0.74 | 0.79 | 0.22 | 0.45 |
| *AR-based method* | | | | | | | |
| LlamaGen (Sun et al., 2024) | 0.32 | 0.71 | 0.34 | 0.21 | 0.58 | 0.07 | 0.04 |
| Emu3 (Wang et al., 2024) | 0.54 | 0.98 | 0.71 | 0.34 | 0.81 | 0.17 | 0.21 |
| Show-o (Xie et al., 2024) | 0.68 | 0.98 | 0.80 | 0.66 | 0.84 | 0.31 | 0.50 |
| GPT-4o (OpenAI, 2025) | 0.85 | 0.99 | 0.92 | 0.85 | 0.91 | 0.75 | 0.66 |
| Janus-Pro-1B (Chen et al., 2025a) | 0.73 | 0.98 | 0.82 | 0.51 | 0.89 | 0.65 | 0.56 |
| Janus-Pro-7B (Chen et al., 2025a) | 0.80 | 0.99 | 0.89 | 0.59 | 0.90 | 0.79 | 0.66 |
| *AR-based Method + RL* | | | | | | | |
| Show-o+PARM (Guo et al., 2025b) | 0.69 | 0.97 | 0.75 | 0.60 | 0.83 | 0.54 | 0.53 |
| T2I-R1 (Jiang et al., 2025) | 0.79 | 0.99 | 0.91 | 0.53 | 0.91 | 0.76 | 0.65 |
| LlamaGen+GRPO | 0.39 | 0.83 | 0.41 | 0.28 | 0.68 | 0.11 | 0.06 |
| Janus-Pro-1B+GRPO | 0.84 | 1.00 | 0.95 | 0.59 | 0.84 | 0.88 | 0.77 |
| Janus-Pro-7B+GRPO | 0.87 | 0.99 | 0.92 | 0.71 | 0.94 | 0.92 | 0.73 |
| *Our* | | | | | | | |
| LlamaGen+GCPO | 0.42 | 0.83 | 0.49 | 0.25 | 0.71 | 0.13 | 0.08 |
| Janus-Pro-1B+GCPO | 0.85 | 1.00 | 0.96 | 0.63 | 0.88 | 0.91 | 0.73 |
| Janus-Pro-7B+GCPO | 0.90 | 0.99 | 0.95 | 0.90 | 0.90 | 0.95 | 0.76 |

Table 2: Comparison results on T2I-CompBench and DrawBench, evaluated by DEQA-Score, ImageReward, and PickScore. The best result is in green . Time: Time consumption of each RL-training step. We achieve state-of-the-art performance without introducing additional time overhead.

| Method | Color↑ | Shape↑ | Texture↑ | Spatial↑ | Non-Spat.↑ | Complex↑ | DEQA↑ | HPS↑ | ImgRwd↑ | PickScore↑ | Time (s)↓ |
|---|---|---|---|---|---|---|---|---|---|---|---|
| *AR-based method* | | | | | | | | | | | |
| LlamaGen (Sun et al., 2024) | 0.4202 | 0.3967 | 0.5103 | 0.0772 | 0.3050 | 0.2908 | 2.70 | 21.62 | -0.36 | 20.27 | - |
| Janus-Pro-1B (Chen et al., 2025a) | 0.3439 | 0.2363 | 0.2788 | 0.0969 | 0.2813 | 0.2733 | 3.38 | 25.46 | -0.19 | 20.79 | - |
| Janus-Pro-7B (Chen et al., 2025a) | 0.6359 | 0.3528 | 0.4936 | 0.2061 | 0.3085 | 0.3559 | 3.55 | 28.00 | 0.68 | 21.82 | - |
| *AR-based Method + RL* | | | | | | | | | | | |
| LlamaGen+GRPO | 0.4454 | 0.4092 | 0.5446 | 0.0780 | 0.3069 | 0.3066 | 2.85 | 25.28 | -0.14 | 20.44 | 7.63 |
| Janus-Pro-1B+GRPO | 0.7050 | 0.3150 | 0.4621 | 0.3020 | 0.2963 | 0.3159 | 3.67 | 29.16 | 0.73 | 21.59 | 9.54 |
| Janus-Pro-7B+GRPO | 0.7478 | 0.3999 | 0.5849 | 0.2481 | 0.3090 | 0.3744 | 3.70 | 30.64 | 0.99 | 22.12 | 14.75 |
| *Our* | | | | | | | | | | | |
| LlamaGen+GCPO | 0.4691 | 0.4351 | 0.5726 | 0.1015 | 0.3086 | 0.3199 | 2.97 | 26.27 | 0.10 | 20.59 | 7.69 |
| Janus-Pro-1B+GCPO | 0.7373 | 0.3201 | 0.4803 | 0.3220 | 0.2948 | 0.3059 | 3.73 | 29.61 | 0.73 | 21.60 | 9.69 |
| Janus-Pro-7B+GCPO | 0.7508 | 0.5173 | 0.7030 | 0.3824 | 0.3133 | 0.3888 | 3.73 | 30.90 | 1.01 | 22.10 | 14.89 |

## 6.2 MAIN RESULTS

We compare our method with leading AR models and diffusion models on GenEval, as shown in Table 1. Only utilizing critical tokens (30% of the total tokens), GCPO achieves significant improvements over GRPO across all three base models. Notably, Janus-Pro-7B+GCPO attained the highest overall score of **0.90**. This is primarily attributed to a substantial improvement in the Counting task (+0.19). Furthermore, Table 2 shows results of our method on T2I-CompBench, which differs substantially from the GenEval-style training data. GCPO consistently outperforms GRPO in the majority of tasks, achieving up to 20% performance gains on Shape (+0.1174), Texture (+0.1181), and Spatial task (+0.1343), thereby demonstrating strong generalization. We analyze the time consumption of our method in Table 2. Our approach achieves the aforementioned performance gains without introducing additional computational overhead (GCPO only increases time consumption by 1%). As shown in Fig. 6, the model with our method accurately understands the shape of the mountain (big) and the tree (small). In contrast, the model with GRPO generates a forest.

As shown in Table 2, under the HPS-based reward setting, our method consistently outperforms GRPO and demonstrates superior performance across all three models. Notably, Janus-Pro-7B+GCPO achieves the best scores on image quality and human preference alignment. Meanwhile, the images generated by our method exhibit more natural and vivid details, as illustrated in Fig. 6. For more comparison results, please refer to Appendix F.

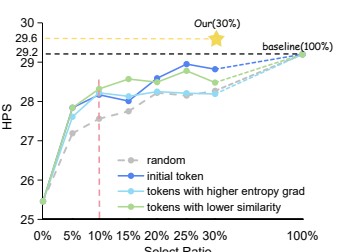 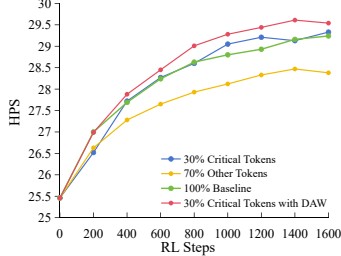 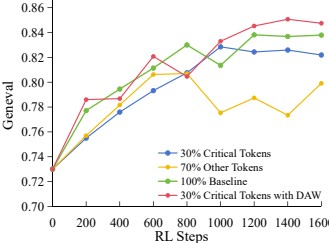

(a) Comparison on token selection strategies and selection ratios.

(b) Comparison of critical tokens and other tokens on HPS

(c) Comparison of critical tokens and other tokens on Geneval

Figure 7: (a): All three types of critical tokens deliver clear performance gains at 10% initial selection ratios and outperform random selection, with further increases yield only limited improvements. (b) & (c): GRPO with critical tokens (30%) has more performance improvement than GRPO with remaining tokens (70%). Our GCPO further achieves performance improvement. We further analyze the settings for the overall selection ratio and the selection ratio for each type in Appendix E.

## 6.3 ABLATION STUDY

We conduct an analysis study on different token-selection strategies and selection rates, as shown in Fig. 7a. Within the initial selection ratios range (10% of tokens), GRPO performance rises rapidly (+2.81) with each of the three types of critical tokens, respectively; beyond this threshold, the gains are limited (+0.95). Moreover, combining all three types of critical tokens clearly outperforms selecting any single type at the same token budget (30% of tokens), which demonstrates the correct identification of critical tokens in RLVL training. We further verify that all three strategies significantly outperform random selection. As shown in Table 3, we present more experimental results on multiple benchmarks under Geneval and HPS reward settings. Our critical token strategies consistently achieve balanced performance improvements. Furthermore, with the introduction of dynamic advantage weight, all metrics reach optimal values, demonstrating the effectiveness of our designs.

Table 3: Ablation results on critical tokens and dynamic advantage weight. Init-T: initial tokens, HG-T: high entropy gradient tokens, LS-T: low similarity tokens; DAW: dynamic advantage weight.

| Init-T | HG-T | LS-T | DAW | GenEval↑ | T2I-CompBench | | | | | | DEQA↑ | HPS↑ | ImgRwd↑ |
| | | | | | Color↑ | Shape↑ | Texture↑ | Spatial↑ | Non-Spat.↑ | Complex↑ | | | |
|---|---|---|---|---|---|---|---|---|---|---|---|---|---|
| ✓ | - | - | - | 0.82 | 0.6084 | 0.2820 | 0.3496 | 0.2374 | 0.2865 | 0.2877 | 3.66 | 28.90 | 0.63 |
| - | ✓ | - | - | 0.81 | 0.5440 | 0.2712 | 0.3368 | 0.2262 | 0.2832 | 0.2794 | 3.63 | 28.22 | 0.63 |
| - | - | ✓ | - | 0.82 | 0.6721 | 0.2942 | 0.4101 | 0.2569 | 0.2938 | 0.3109 | 3.64 | 28.78 | 0.66 |
| ✓ | ✓ | ✓ | - | 0.83 | 0.6602 | 0.2924 | 0.3991 | 0.2936 | 0.2943 | 0.3015 | 3.70 | 29.33 | 0.71 |
| ✓ | ✓ | ✓ | ✓ | 0.85 | 0.7373 | 0.3201 | 0.4803 | 0.3220 | 0.2948 | 0.3059 | 3.73 | 29.61 | 0.73 |

We also consider the comparison between 30% critical tokens and 70% of the remaining tokens. As shown in Fig. 7b and 7c, GRPO with critical tokens are comparable with the GRPO baseline, while GRPO with other tokens leads to a drop in performance. Despite the fact that the other tokens account for 70% of the total tokens used in training and more than twice the count of critical tokens, they still have a significant performance gap. This finding highlights the greater importance of critical tokens based on our selection strategy for effective model training.

In addition, to further illustrate the rationality of the selection ratio for each type and the overall selection ratio, we provide more experiments in the Appendix E. As shown in Table. 8, we observe that the imbalanced selection for each type significantly degrades performance compared to balanced selection (10%:10%:10%), which is also supported by the experimental results in Fig. 7a.

Notably, increasing the total selection ratios from 30% to 45% yields almost no performance improvement, but decreasing it from 30% to 15% results in a significant performance degradation, as shown in the Table. 10. Furthermore, increasing total selection ratios to the entire sequence of tokens leads to a decline in performance. This phenomenon arises from the use of DAW on the entire sequence of tokens, for which we conduct additional experiments and analyses in the Appendix E.4.

Next Token Prediction (NTP) is currently the most general paradigm in NLP and AR image generation. Nevertheless, based on the intrinsic spatial structure of images, previous works (Tian et al., 2024; Han et al., 2025; Ma et al., 2024) have explored next-scale prediction to achieve high-quality and efficient image generation. We successfully extend our method to the next-scale prediction generation paradigm. Specifically, we expand our method as follows: the importance of initial tokens (AR causality) is extended to the importance of early scales, while entropy-gradient-based structure and similarity-based diversity mechanisms can be seamlessly adapted. Our approach demonstrates strong generalizability and achieves superior performance to GRPO across multiple metrics. This suggests that the key concept of critical tokens is general and worthy of further investigation. **Notably, our GCPO utilizes only critical tokens (25% of the tokens).** The comparison results are shown in Table. 4 and Fig. 14.

Table 4: Results on scale-wise model. The best result is in green .

| Method | DEQA↑ | HPS↑ | ImgReward↑ | PickScore↑ | GenEval↑ |
|---|---|---|---|---|---|
| Star (Ma et al., 2024) | 3.70 | 26.38 | 0.52 | 21.84 | 0.47 |
| Star (Ma et al., 2024) + GRPO | 4.11 | 30.17 | 0.76 | 22.19 | 0.50 |
| Star (Ma et al., 2024) + GCPO (our) | 4.15 | 30.32 | 0.78 | 22.17 | 0.49 |

## 7 CONCLUSION

In this paper, we introduce Group Critical-token Policy Optimization (**GCPO**), a novel RLVR framework for autoregressive image generation. We identify critical tokens in RLVR-based AR visual generation from three perspectives: Causal dependency of AR, Entropy-induced spatial structure, and RLVR-focused token diversity. We devise the dynamic advantage weight for critical tokens to enable reasonable exploration, based on their confidence divergence between the policy model and reference model. Extensive experiments demonstrate that by leveraging critical tokens (30% of the image tokens), GCPO achieves better performance than GRPO, which operates on the full tokens.

## 8 ACKNOWLEDGMENTS

This work was supported by the Anhui Provincial Natural Science Foundation under Grant 2108085UD12. We acknowledge the support of GPU cluster built by MCC Lab of Information Science and Technology Institution, USTC. The AI-driven experiments, simulations and model training were performed on the robotic AI-Scientist platform of Chinese Academy of Sciences.

## 9 ETHICS STATEMENT

**Data Usage.** All training and evaluation datasets used in this study are publicly available and have been widely adopted in prior research, including Geneval, T2I-CompBench, DrawBench, and HPSv2. We do not collect or distribute any new human or proprietary data. The prompts and images used do not contain personally identifiable information or sensitive content.

**Content and Bias.** We do not introduce any new annotations or external knowledge sources that could inject additional biases beyond those already present in the original models and datasets. To further promote responsible use, we encourage the integration of existing safety filters, content-moderation tools, and bias-detection techniques when deploying models enhanced with our method.

## 10 REPRODUCIBILITY STATEMENT

Here, we provide content for better Reproducibility.

1. The detailed method is described in Sec. 5
2. Hyperparameters for all used models in this paper (see Sec. A).

Besides, all training data, checkpoints, and code will be released within the scope of the conference.

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

## A  DETAILED EXPERIMENTAL SETUP

### A.1  HYPERPARAMETER

In this paper, we used three models (LlamaGen (Sun et al., 2024), Janus-Pro 1B (Chen et al., 2025a), and Janus-Pro 7B (Chen et al., 2025a)) on the composition image task and image quality task. For different models and reward (HPS Reward (Wu et al., 2023) and Geneval Reward (Liu et al., 2025))settings, our training configuration and parameters are as follows:

Table 5: GCPO training hyperparameters.

| Name | LlamaGen for HPS | 1B for Geneval | 1B for HPS | 7B for Geneval | 7B for HPS |
|---|---|---|---|---|---|
| Learning rate | 1e-5 | 3e-6 | 1e-6 | 3e-6 | 1e-6 |
| Beta $\beta$ | 0.04 | 0.01 | 0.01 | 0.01 | 0.01 |
| Group Size $G$ | 8 | 4 | 4 | 4 | 8 |
| Classifier-Free Guidance Scale | 1 | 5 | 5 | 5 | 5 |
| Max Gradient Norm | 1.0 | 1.0 | 1.0 | 1.0 | 1.0 |
| Batchsize | 2 | 2 | 2 | 2 | 4 |
| Training Steps | 900 | 1,600 | 1,600 | 1,600 | 1,600 |
| Gradient Accumulation Steps | 2 | 1 | 1 | 1 | 1 |
| Dynamic advantage weight clip $\epsilon_w$ | 0.5 | 1 | 0.5 | 0.6 | 1 |
| Image Resolution $h \times w$ | $256 \times 256$ | $384 \times 384$ | $384 \times 384$ | $384 \times 384$ | $384 \times 384$ |

### A.2  TRAINING EFFICIENCY

Compared with standard GRPO, our additional operations involve entropy gradient calculation and token embedding similarity computation. We have implemented certain optimizations, so these operations do not introduce additional computational burden or affect training efficiency.

## B  ENTROPY ANALYSIS IN RLVR-BASED AR VISUAL GENERATION

In section 4.2, we discuss the entropy distribution of images in AR generation. We observe that the entropy distribution of image tokens exhibits a spatial pattern and fails to maintain consistency in different prompts. We further provide some examples and statistical results to illustrate this point, as shown in Fig. 8

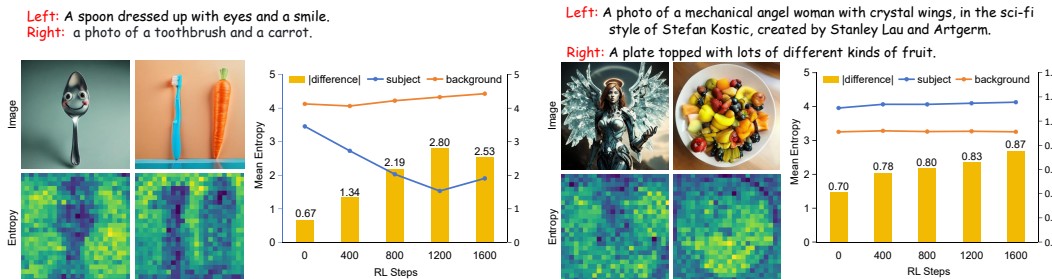

(a) The entropy distribution of simple or compositional prompts. The entropy of the background is significantly greater than that of the subject.

(b) The entropy distribution of complex or detailed prompts. The entropy of the subject is greater than that of the background.

Figure 8: In each figure, the blue line represents the average entropy of the subject regions, while the orange line represents the average entropy of the background regions. |difference| represents the average entropy difference between subject and background regions.

We argue that this entropy distribution is directly related to the complexity of the prompts. Specifically, for composition prompts, such as Geneval-style prompts, the entropy of the subject region is lower than that of the background region (see Fig. 8a). These prompts are relatively simple, including only the necessary keywords without excessive modifiers. As a result, the model has higher uncertainty when generating tokens in the background region due to the lack of sufficient prompt

information. For complex prompts, such as the HPS testing data, the subject regions have a higher information density, resulting in higher entropy for tokens in these regions (see Fig. 8b).

Based on this insight, we conduct corresponding experiments to verify this point. We analyze the images generated by Janus-Pro and the corresponding entropy distribution, where prompts come from the Geneval Benchmark and HPS testing data. We use RMBG v2.0 (briaai, 2025), a state-of-the-art background removal model, to separate the foreground and background regions of the generated images. Then, we calculate the average token entropy within the foreground region, as shown in Fig. 8. The statistical results further verify our view.

In addition, we further investigate the evolution of entropy distribution during the RL training process. We find that RL training does not alter the original entropy pattern, but instead further reinforces it, which is similar to observations in LLMs (Li et al., 2025; Vassoyan et al., 2025). This phenomenon means that the entropy difference between the subject and background regions gradually increases (see the histogram in Fig. 8), resulting in more pronounced entropy (information) changes for high-gradient tokens in their neighborhood. Such information variation and complex structural regions are simultaneously coupled to these tokens, prompting us to include them in the scope of critical tokens for focused optimization.

## C MORE COMPARISON RESULTS WITH RELATED WORK

In the field of AR image generation based on RLVR, (Jiang et al., 2025) and (Pan et al., 2025a) are representative works that are close to ours. The differences are that (Jiang et al., 2025) introduces semantic-CoT before image generation and uses a combination of multiple reward models, including HPS (Wu et al., 2023), GroundingDINO (Liu et al., 2024b), GIT (Wang et al., 2022), and LLaVA-OneVision-7B (Li et al., 2024), and constructs training data that includes both Geneval-style and T2i-compbench-style prompts. We only utilize Geneval rewards and Geneval-style data. (Pan et al., 2025a) constructs fine-grained paired prompt-image training data and first trains the model on paired data with images, then operates GRPO training without images. In contrast, we only perform image-free RL training. Comparison results on Geneval and T2i-compbench are shown in the Table 6. Our method achieves a significant lead on Geneval and also obtains comparable results on T2i-compbench.

Table 6: Comparison results with T2I-R1 and Focus-Diff on Geneval and T2I-CompBench.

| Method↑ | GenEval↑ | T2I-CompBench | | | | | |
| --- | --- | --- | --- | --- | --- | --- | --- |
| | | Color↑ | Shape↑ | Texture↑ | Spatial↑ | Non-Spat.↑ | Complex↑ |
| T2I-R1 (Jiang et al., 2025) | 0.79 | 0.8130 | 0.5852 | 0.7243 | 0.3378 | 0.3090 | 0.3993 |
| Focus-Diff (Pan et al., 2025a) | 0.85 | 0.7996 | 0.5748 | 0.7007 | 0.3789 | 0.3098 | 0.3912 |
| Janus-Pro-7B+GRPO | 0.87 | 0.7478 | 0.3999 | 0.5849 | 0.2481 | 0.3090 | 0.3744 |
| Janus-Pro-7B+GCPO | 0.90 | 0.7508 | 0.5173 | 0.7030 | 0.3824 | 0.3133 | 0.3888 |

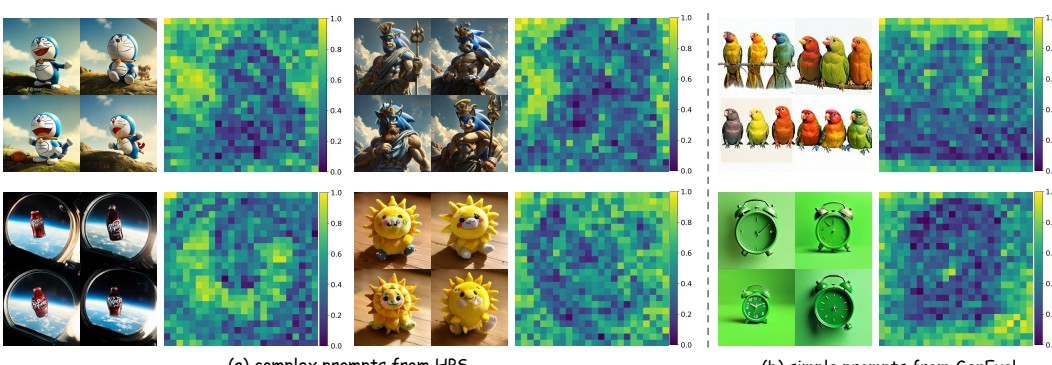

(a) complex prompts from HPS  (b) simple prompts from GenEval

Figure 9: Visual results from different benchmarks

# D  ANALYSIS BETWEEN SIMILARITY AND REGIONS IN AR GENERATION

We provide more examples from the HPS benchmark and GenEval benchmark to further illustrate the relationship between similarity and regions, as shown in the Fig. 9. It is obvious that simple background and texture regions in images within the same group tend to be more consistent (higher similarity), while complex regions show greater diversity (lower similarity).

# E  ADDITIONAL EXPERIMENTS

## E.1  THE INFLUENCE OF DYNAMIC ADVANTAGE WEIGHT

Dynamic Advantage Weight (DAW) is designed to work in cooperation with the critical token strategy and improve performance by further encouraging diverse advantages among the critical tokens. Applying DAW to all the 100% tokens may amplify certain negative gradients among the remaining non-critical tokens. To verify this, we conduct the corresponding experiments, as shown in Tabel. 7.

As shown in Table. 7, the experimental results show that our method (30% tokens with DAW) > (100% tokens with DAW) > (100% tokens without DAW). This verifies that 1) DAW is also useful to the general GRPO. 2) Critical tokens and DAW work synergistically to achieve the best performance. We argue that DAW may amplify certain negative gradients among the remaining non-critical tokens. Therefore, the rational selection of critical tokens is essential for performance improvement, and combining this selection with DAW leads to even better results.

Table 7: Ablation results on dynamic advantage weight

| Method | DEQA↑ | HPS↑ | ImgReward↑ | PickScore↑ | GenEval↑ |
|---|---|---|---|---|---|
| 100% Tokens without DAW (GRPO) | 3.67 | 29.16 | 0.73 | 21.59 | 0.84 |
| 100% Tokens with DAW | 3.71 | 29.54 | 0.72 | 21.62 | 0.85 |
| 30% Critical Tokens without DAW | 3.70 | 29.33 | 0.71 | 21.58 | 0.83 |
| 30% Critical Tokens with DAW (our) | 3.73 | 29.61 | 0.73 | 21.60 | 0.85 |

## E.2  RESULTS ON ADDITIONAL SELECTION RATIO SETTINGS

We further provide experiments with other selection ratios, as shown in Table 8. We observe that when the total selection ratio is 30%, and the selection ratio is set to 15%:10%:5%, the performance decreases significantly even though the total number of selected tokens remains unchanged.

We compare multiple ratios for each type of critical token, as shown in Fig. 7. For each type, 10% serves as a performance gain threshold. Beyond this point, performance improves slowly. Therefore, we adopt 10% as the default for each type.

Table 8: Comparison results on additional selection ratio settings

| Method | DEQA↑ | HPS↑ | ImgReward↑ | PickScore↑ | GenEval↑ |
|---|---|---|---|---|---|
| 30% (15%:10%:5%) | 3.66 | 29.54 | 0.62 | 21.54 | 0.83 |
| 30% (10%:10%:10%) (our) | 3.73 | 29.61 | 0.73 | 21.60 | 0.85 |

## E.3  THE ABLATION ON LOCAL AVERAGING OF THE ENTROPY MAP

Local averaging of the entropy map is proposed to reduce the noise and outliers in the entropy map, enabling more robust and efficient critical token selection. We further provide corresponding ablation experiments on this strategy in the Table. 9, where removing the local averaging results in a slight decrease in model performance.

Table 9: Comparison results on local averaging

| Method | DEQA↑ | HPS↑ | ImgReward↑ | PickScore↑ | GenEval↑ |
|---|---|---|---|---|---|
| w/o local averaging | 3.71 | 29.56 | 0.71 | 21.58 | 0.84 |
| w local averaging (our) | 3.73 | 29.61 | 0.73 | 21.60 | 0.85 |

### E.4 THE ABLATION ON THE TOTAL SELECTION PROPORTION

In Table. 10, We provide experiments on the total selection proportion, including 15% (5%+5%+5%) and 45% (15%+15%+15%). We observe that reducing the selection proportion to 15% results in a significant performance drop, while further increasing it does not bring notable gains. Therefore, we outperform the GRPO with all tokens using as few critical tokens as possible.

Table 10: Ablation results on total selection proportion

| Method | DEQA↑ | HPS↑ | ImgReward↑ | PickScore↑ | GenEval↑ |
|---|---|---|---|---|---|
| 15% (5%+5%+5%) | 3.63 | 29.29 | 0.63 | 21.55 | 0.82 |
| 30% (10%+10%+10%) (our) | 3.73 | 29.61 | 0.73 | 21.60 | 0.85 |
| 45% (15%+15%+15%) | 3.70 | 29.65 | 0.75 | 21.62 | 0.85 |

### E.5 THE INFLUENCE OF GROUP SIZE

We argue that for the GRPO algorithm itself, a larger group size is empirically verified to yield more stable policy variance and a more diverse exploration space. Brorl (Hu et al., 2025) points out that substantially increasing the group size of the LLM can further improve the performance of models that are experiencing learning stagnation. Previous methods (Jiang et al., 2025; Zhang et al., 2025) typically set group size as a trade-off between performance and memory requirements.

To further validate this on AR visual generation, we provide more experiments of our method on different group sizes as shown in Table 11. We find that group size and HPS have a strong positive correlation. DEQA, PickScore, and ImageReward show significant improvements from 4 to 8, but the improvement is minimal when the group size is further increased (from 8 to 12). Conversely, the GenEval only shows an improvement when the group size increases (from 8 to 12).

Table 11: Comparison results on different group sizes

| Method | GPU Memory↓ | Training Time↓ | DEQA↑ | HPS↑ | ImgReward↑ | PickScore↑ | Geneval↑ |
|---|---|---|---|---|---|---|---|
| G=4 | 39G | 4.5h | 3.73 | 29.61 | 0.73 | 21.60 | 0.85 |
| G=8 | 79G | 9.2h | 3.80 | 30.39 | 0.82 | 21.70 | 0.86 |
| G=12 | 115G | 11.4h | 3.81 | 30.63 | 0.85 | 21.72 | 0.88 |

## F MORE VISUAL COMPARISON RESULTS

In this section, we further provide visual comparison results on the composition image task and image quality task to better illustrate the effectiveness and compatibility of our method. We will provide prompts for all images in Sec. F.5.

### F.1 COMPARISON ON LLAMAGEN

We only use the HPS Reward on LlamaGen and abandon the Geneval reward. This is because we find that the Geneval score for each image generated by LlamaGen is close to 0 and does not provide effective training rewards. Nevertheless, the model trained with the HPS reward not only demonstrates better generation quality, as shown in 10, but also further improves performance on Geneval.

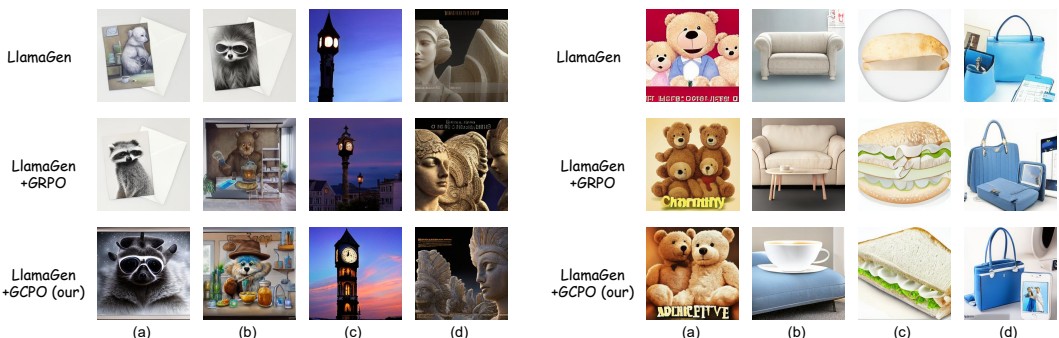

(a) Visual comparison results on HPS benchmark.   (b) Visual comparison results on Geneval

Figure 10: Qualitative comparison between the LlamaGen + GRPO and LlamaGen + GCPO (our) trained with HPS reward. Prompts are in Sec. F.5

## F.2 COMPARISON ON JANUS-PRO 1B

We provide more qualitative comparison results on Janus-Pro 1B, as shown in 11 (HPS reward) and 12 (Geneval reward). These visual results further demonstrate the effectiveness of our approach.

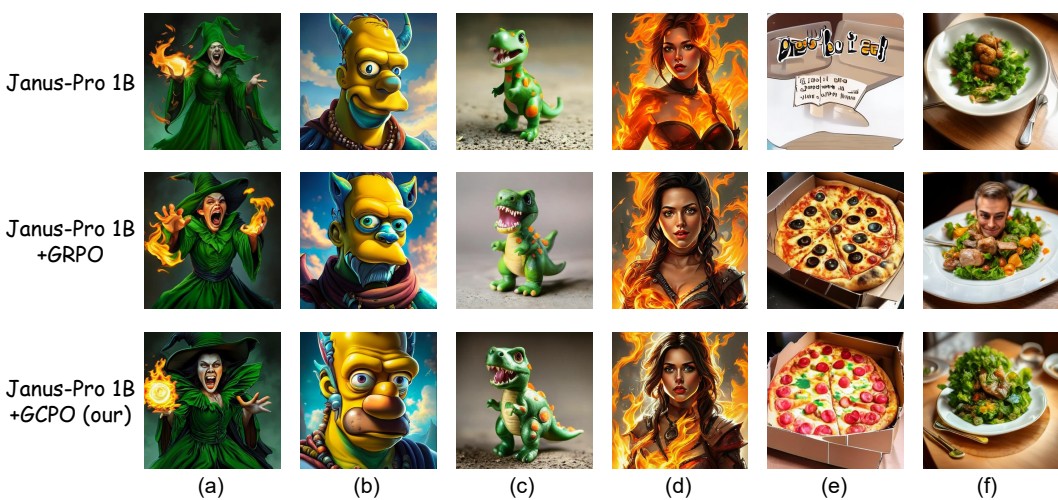

Figure 11: Qualitative comparison between the Janus-Pro 1B + GRPO and Janus-Pro 1B + GCPO (our) trained with HPS reward. Prompts are in Sec. F.5

## F.3 MORE VISUAL RESULTS DURING RLVR TRAINING

To better understand the training dynamics of our method, we visualize the results of samples generated by the same prompts during training, as shown in Fig. 13. These qualitative results intuitively demonstrate how the model continuously optimizes towards the goal of improving image quality and Human Preference Alignment as training progresses.

## F.4 COMPARISON ON STAR

We provide more qualitative comparison results on the scale-wise model, Star (Ma et al., 2024), as shown in Fig. 14. These visual results further demonstrate the extension of our approach.

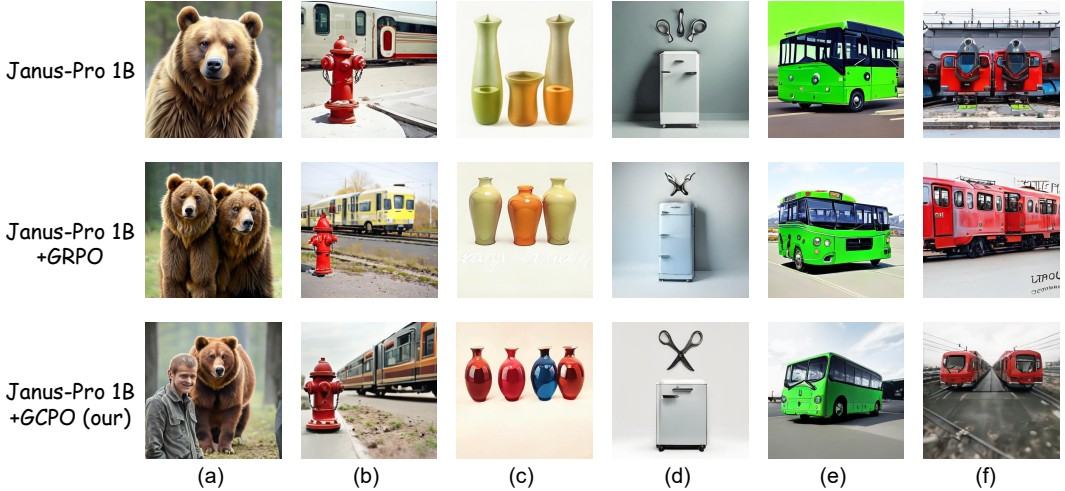

Figure 12: Qualitative comparison between the Janus-Pro 1B + GRPO and Janus-Pro 1B + GCPO (our) trained with Geneval reward. Prompts are in Sec. F.5

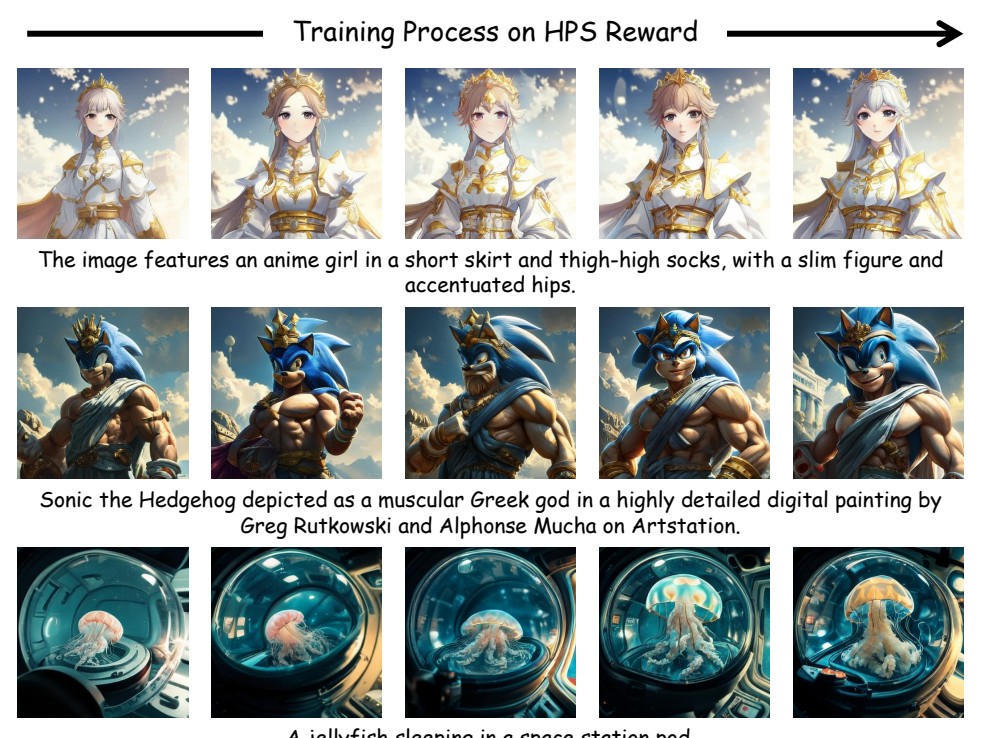

Figure 13: We visualize the generated samples during the optimization of Janus-Pro 7B trained with HPS reward. As training progressed, the model is steadily optimized towards the target of Human Preference Alignment.

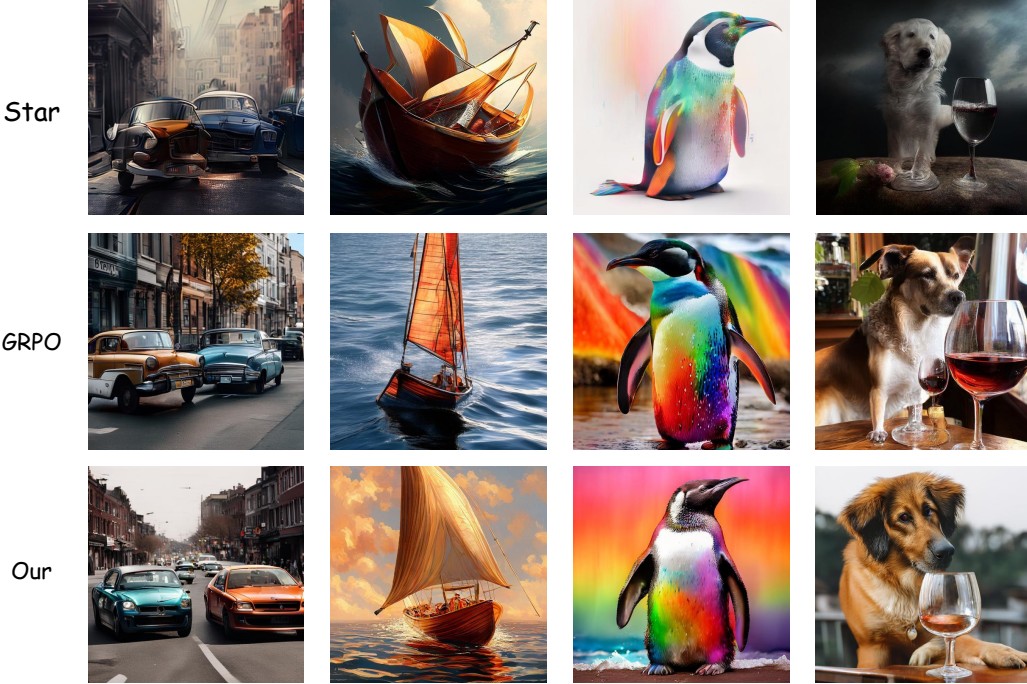

Figure 14: Visualization Results on Star.

### F.5 USED PROMPTS IN THIS SECTION

The prompts used in Fig. 10a are:

(a) The image is of a raccoon wearing a Peaky Blinders hat, surrounded by swirling mist and rendered with fine detail.

(b) A teddy bear mad scientist mixing chemicals depicted in oil painting style as a fantasy concept art piece.

(c) A clock tower with lighted clock faces, against a twilight sky.

(d) The image features a closeup portrait of stone angel statues, created with the Unreal Engine and featuring intricate details by various artists.

The prompts used in Fig. 10b are:

(a) a photo of two teddy bears

(b) a photo of a couch below a cup

(c) a photo of a white sandwich

(d) a photo of a blue handbag and a white cell phone

The prompts used in Fig. 11 are:

(a) Wicked witch casting fireball dressed in green with screaming expression.

(b) The image is a portrait of Homer Simpson as a Na'vi from Avatar, created with vibrant colors and highly detailed in a cinematic style reminiscent of romanticism by Eugene de Blaas and Ross Tran, available on Artstation with credits to Greg Rutkowski.

(c) A small green dinosaur toy with orange spots standing on its hind legs and roaring with its mouth open.

(d) Mila Kunis portrayed as a fire elemental in a highly detailed digital painting.

(e) A pizza is displayed inside a pizza box.

(f) A portrait of a dinner dish of a protein and greens.

The prompts used in Fig. 12 are:

(a) a photo of a person and a bear

(b) a photo of a fire hydrant and a train

(c) a photo of four bowls

(d) a photo of a baseball glove right of a bear

(e) a photo of a green bus

(f) a photo of two trains

## G THE USE OF LARGE LANGUAGE MODELS (LLMS)

In this paper, LLMs are employed to refine the writing, further enhancing the readability and quality of the paper.

