# OpenReview forum: "Group Critical-token Policy Optimization for Autoregressive Image Generation"
_ICLR.cc/2026/Conference — ICLR 2026 Poster_

### Official Review · Reviewer_eipe · 2025-10-20

**Soundness:** 3
**Presentation:** 3
**Contribution:** 2
**Rating:** 6
**Confidence:** 4

**Summary:**

This papers proposes Group Critical-token Policy Optimization for autoregressive image generation models drawing insights from Reinforcement Learning with Verifiable Rewards in the context of LLMs and Chain of Thought reasoning.

**Strengths:**

- The paper identifies a more efficient and insightful method to improve RL post-training for AR models.
- The paper shows good empirical performance
- The paper carries out extensive experiments

**Weaknesses:**

The paper does not discuss the recent Visual Autoregressive VAR paradigm: https://arxiv.org/pdf/2404.02905
There are works that have used Group Relative Policy Optimization for VAR such as 'Fine-Tuning Next-Scale Visual Autoregressive Models with Group Relative Policy Optimization': https://arxiv.org/abs/2505.23331

These should be discussed in the text, and extensions of GCPO to VAR should be mentioned as a future research direction. The problem with standard AR is that they impose a raster-scan inductive bias on image generation which does not align with the multi-scale nature of images: there is no reason to believe that the top left corner should condition the bottom right corner and not the other way around. As the authors mention 'early tokens serve as global priors and structural guides', which more closely aligns with a multi-scale approach.

**Questions:**

The paper relies on heuristics which may make the method brittle and sensitive to hyper-parameters:

- The assumption is that the first K tokens (e.g., 10% of the sequence length ) are critical because their initial generation decisions propagate and determine the overall image structure (why this particular number?)
- The 2D gradient of the token entropy map. The authors observed empirically that tokens with high entropy gradients consistently correspond to important image structures and region-bridging elements. This is interesting, any theoretical insights or links to grad-cam and interpretability methods?
- Token diversity within a group of generated images. The criterion selects tokens with the lowest average pairwise cosine similarity of their embeddings across a batch of images. The rationale is that low-similarity tokens are associated with more complex regional structures that contribute richer information for policy optimization, while high-similarity tokens are mundane background or texture. Any further experiments to check this? Perhaps and appendix with more examples and images.

---

> ### Author Response · Authors · 2025-11-21
> **Response to Reviewer eipe**
>
> We sincerely thank the reviewer for the positive assessment of our work and insightful comments. We hope our response below can address the reviewer's concerns.
>
> > **W1:** The paper does not discuss the recent Visual Autoregressive VAR paradigm, and extensions of GCPO to VAR should be mentioned as a future research direction.
>
> Thanks for your valuable question and discussion, and we also agree that the raster-scan AR itself may not be optimal for image generation. Therefore, following your suggestion, we have supplemented the discussion of VAR and GRPO for VAR in the revised version (**see line 119**), and consider the extension of GCPO as a future research direction. It is worth noting that we have **already achieved promising results on a scale-wise model (Star [1])**, which demonstrates the potential and generalization of our method. As shown in **Table 4** and the following table. Notably, our method, using **only 25% of the tokens**, still outperforms GRPO on multiple metrics, which is trained on full sequence tokens. We provide visual results in **Figure 14**.
>
> | Method                           | DEQA↑ | HPS↑  | ImgReward↑ | PickScore↑ | GenEval↑ |
> |:----------------------------------|:-----:|:-----:|:----------:|:----------:|:--------:|
> | Star          | 3.70  | 26.38 | 0.52       | 21.84      | 0.47     |
> | Star + GRPO    | 4.11  | 30.17 | 0.76       | **22.19**      | **0.50**     |
> | Star + GCPO (our) | **4.15**| **30.32** | **0.78**       | 22.17      | 0.49     |
>
> We have added a more detailed analysis and discussion of the scale-wise model in **Section 6.4** of the revised manuscript.
>
> > **Q1:** The assumption is that the first K tokens (e.g., 10% of the sequence length ) are critical because their initial generation decisions propagate and determine the overall image structure (why this particular number?)
>
> This selection ratio comes from our experimental observations. We conduct an ablation analysis of initial token selection ratios, as shown in **Figure 7**. The performance improves significantly as the proportion of selected early tokens increases up to 10%, but the improvement slows beyond this point. Thus, we identify 10% as the optimal proportion for the initial critical tokens.
>
> >**Q2:** The authors observed empirically that tokens with high entropy gradients consistently correspond to important image structures and region-bridging elements. This is interesting. Any theoretical insights or links to grad-cam and interpretability methods?
>
> Thank you for acknowledging our perspective on entropy gradients. The role of tokens with high entropy gradient originates from our experimental observations. We provide statistical analyses of 553 and 400 prompts in **Appendix B**, showing a significant difference in entropy between the foreground and background regions; these tokens with high entropy gradient tend to correspond to these boundary regions.
>
> During the research process, we also noticed these two papers: GradEn [2] and Slope Entropy [3]. They both combine entropy and local gradients to analyze one-dimensional data and two-dimensional images, respectively. Although our method differs from theirs, this approach of combining entropy with information changes has proven feasible.
>
> >**Q3:** The rationale is that low-similarity tokens are associated with more complex regional structures that contribute richer information for policy optimization, while high-similarity tokens are mundane background or texture. Any further experiments to check this? Perhaps an appendix with more examples and images.
>
> We provide more examples in **Appendix D** to illustrate that low-similarity tokens are often associated with more complex regional structures, while high-similarity tokens tend to be background or texture. These cases provide a more direct understanding of the relationship between similarity and the complexity of regions.
>
> >**Reference**
>
> [1] STAR: Scale-wise Text-conditioned AutoRegressive image generation
>
> [2] Gradient entropy (GradEn): The two-dimensional version of slope entropy for image analysis
>
> [3] Slope Entropy: A New Time Series Complexity Estimator Based on Both Symbolic Patterns and Amplitude Information

---

> ### Author Response · Authors · 2025-11-28
> **Welcome for Further Feedback**
>
> Dear Reviewer eipe,
>
> We sincerely thank you for your thorough and constructive feedback, which greatly guided our additional analyses during the rebuttal period. Following your insightful comments, we have conducted new experiments and ablation studies as presented in our **Rebuttal**. These updates have been incorporated into the revised manuscript and appendix to more clearly address each of your points.
>
> - **Regarding more generative paradigm:** we explore combining a scaling model (Star) with our method (**using** **only 25% of the tokens**) and have achieved promising results. These results are included in **Section 6.4** and **Figure 14**.
> - **More explanations:** we further provide details and clarifications to help you better understand certain aspects of our method.
>
> We hope that these substantial revisions fully address your thoughtful questions and concerns. If there are any remaining issues that require further clarification, we would be grateful for the opportunity to continue the discussion. Thank you again for your time, expertise, and constructive guidance.

---

### Official Review · Reviewer_KyYg · 2025-10-31

**Soundness:** 3
**Presentation:** 3
**Contribution:** 3
**Rating:** 6
**Confidence:** 3

**Summary:**

This paper introduces Group Critical-token Policy Optimization (GCPO), a novel RLVR framework that focuses on optimizing critical tokens, regarding to causal dependency, structure importance, and token diversity, for autoregressive image generation, achieving significant improvements in efficiency and performance over GRPO across benchmarks.

**Strengths:**

* Novelty: Introduces a critical-token selection strategy based on causal dependency, entropy gradients, and token diversity.
* Exploration: Provides intriguing insights into the role of structure in token entropy during text-to-image generation.
* Comprehensive Validation: Demonstrates improvements across Geneval, T2I-CompBench and DrawBench benchmarks.

**Weaknesses:**

* Limited Gains: Improvements over GRPO are minor, especially for Geneval on 1B models and T2I-CompBench on 7B models, despite added complexity.

**Questions:**

1. Given the limited performance gains, could the authors clarify additional advantages, such as the impact on RL training time due to token filtering?
2. How would GCPO perform on non-spatial vision tokenizers like VAR’s application on T2I [1] and DDT [2] for vision token extraction?

[1] Infinity∞: Scaling Bitwise AutoRegressive Modeling for High-Resolution Image Synthesis

[2] Generative Multimodal Pretraining with Discrete Diffusion Timestep Tokens

---

> ### Author Response · Authors · 2025-11-21
> **Response to Reviewer KyYg**
>
> We sincerely thank the reviewer for the positive assessment of our work and insightful comments. We hope our response below can address the reviewer's concerns.
>
> > **W1:** Limited Gains: Improvements over GRPO are minor, especially for Geneval on 1B models and T2I-CompBench on 7B models, despite added complexity.
>
> We respectfully note that 7B with GCPO achieves a significant improvement over GRPO on T2I-CompBench (e.g., **shape: 0.399→0.5173, texture: 0.5849→0.7030, spatial: 0.2481→0.3824**), and these performance gains are quite considerable, as shown in **Table 2**. Furthermore, we would like to clarify the significance of our work:
>
> (1) Besides performance gains, we want to emphasize more that our method, **for the first time**, reveals the unique perspective of critical tokens in the RL of autoregressive image generation. We identify and validate critical tokens in RL-based AR visual generation from three perspectives, providing valuable insights for further improving RL post-training for AR models.
>
> (2) Our experiments demonstrate that GCPO training based on these **critical tokens (30% of the tokens)** outperforms GRPO training on 100% of the tokens across multiple models and benchmarks. This is pretty challenging and non-trivial.
>
> (3) Our method is simple but effective in implementation, **without additional computational overhead**. We additionally calculate the computation time for GRPO and our method in a step.
> | Method                   | time  |
> |--------------------------|-------|
> | 1B+GRPO  |   9.54s  |
> | 1B+our |  9.54s + 0.15s  |
>
> We have added other models' time consumption of each training step in **Table 2** of the revised manuscript.
>
> > **Q1:** Could the authors clarify additional advantages, such as the impact on RL training time due to token filtering?
>
> (1) The main time consumption in the RL training lies in policy model sampling and reward model scoring. Our method does not introduce any additional computational overhead and therefore does not have any negative impact on training time.
>
> (2) Our method achieves the same performance as GRPO using only 75% of the steps required by GRPO, as shown in **Figure 7**.
>
> (3) Notably, our method's most significant contribution is providing a valuable approach to further overcome the performance bottleneck of RL in AR visual generation.
>
> > **Q2:**  How would GCPO perform on non-spatial vision tokenizers like VAR’s application on T2I and DDT for vision token extraction?
>
> Thank you for pointing out this valuable research direction. Firstly, we believe that Next Token Prediction (NTP) is currently the most general paradigm in NLP and AR visual generation. It still attracts intensive attention from both the research and industry communities. The core of our method is designed for the general paradigm and validates its effectiveness on multiple models. Just as VAR and DDT are innovative works exploring new paradigms in the field of vision, following your suggestion, we have supplemented the discussion of DDT, VAR, and GRPO for VAR in the revised version (**see line 94 and line 119**), and consider the extension of GCPO as a future research direction.
>
> To further illustrate the universality of our method, we explore combining a scaling model (Star [1]) with our method and have achieved promising results, as shown in **Table 4** and the following table. Notably, our method, using **only 25% of the tokens**, still outperforms GRPO on multiple metrics, which is trained using full sequence tokens. We provide more details and visual results in **Section 6.4** and **Figure 14**.
>
> | Method                           | DEQA↑ | HPS↑  | ImgReward↑ | PickScore↑ | GenEval↑ |
> |:----------------------------------|:-----:|:-----:|:----------:|:----------:|:--------:|
> | Star          | 3.70  | 26.38 | 0.52       | 21.84      | 0.47     |
> | Star + GRPO    | 4.11  | 30.17 | 0.76       | **22.19**      | **0.50**     |
> | Star + GCPO (our) | **4.15**| **30.32** | **0.78**       | 22.17      | 0.49     |
>
> We have added a more detailed analysis and discussion of the scale-wise model in **Section 6.4** of the revised manuscript.
>
> >**Reference**
>
> [1] STAR: Scale-wise Text-conditioned AutoRegressive image generation

---

> ### Author Response · Authors · 2025-11-28
> **Welcome for Further Feedback**
>
> Dear Reviewer KyYg,
>
> We sincerely thank you for your thorough and constructive feedback, which greatly guided our additional analyses during the rebuttal period. Following your insightful comments, we have conducted new experiments and ablation studies as presented in our **Rebuttal**. These updates have been incorporated into the revised manuscript and appendix to more clearly address each of your points.
>
> - **Regarding performance gains:** we further clarify the significance of our work and considerable performance gains. We believe that our work provide valuable insights for further improving RL post-training for AR models.
> - **Regarding more generative paradigm:** we explore combining a scaling model (Star) with our method (**using** **only 25% of the tokens**) and have achieved promising results. And we consider the extension of GCPO as a future research direction, following your constructive feedback. These results are included in **Section 6.4** and **Figure 14**.
>
> We hope that these substantial revisions fully address your thoughtful questions and concerns. If there are any remaining issues that require further clarification, we would be grateful for the opportunity to continue the discussion. Thank you again for your time, expertise, and constructive guidance.

---

### Official Review · Reviewer_z8AU · 2025-11-01

**Soundness:** 2
**Presentation:** 2
**Contribution:** 3
**Rating:** 6
**Confidence:** 4

**Summary:**

Current RL methodologies typically assign similar weights to all image tokens, but different tokens are expected to play different roles. To distinguish among tokens, the authors identify three types of critical tokens based on the causal nature of autoregressive models, token entropy, and token diversity. In addition, they apply differentiated weights by comparing each identified token with the initialization model.

**Strengths:**

- Considering that different image tokens play different roles in image generation, it is reasonable to focus on optimizing critical image tokens using RL.
- The paper provides the valuable insight that image tokens with high entropy gradients convey spatial structural information of the generated images.
- The proposed RL method demonstrates better performance than GRPO on both Janus-Pro and LLaMaGen models.

**Weaknesses:**

- Since critical tokens can change during training, they should be re-identified at each training step, which may introduce substantial additional computation. However, the paper lacks a comparison of training cost/consumption.
- The performance of the 1B model drops on the GenEval benchmark in Table 1. What is the reason?
- Figure 7(a) only shows the effect of the selection ratio for each token type, while the analysis does not address the choice of the total selection proportion (e.g., 30%).

**Questions:**

Since critical tokens may overlap across the defined categories, does this situation require additional processing?

---

> ### Author Response · Authors · 2025-11-21
> **Response to Reviewer z8AU**
>
> We sincerely thank the reviewer for the positive assessment of our work and insightful comments. We hope our response below can address the reviewer's concerns.
>
> > **W1:** Since critical tokens can change during training, they should be re-identified at each training step, which may introduce substantial additional computation. However, the paper lacks a comparison of training cost/consumption.
>
> Thank you for providing useful feedback. Although the critical tokens change dynamically during training, our selection operation is **simple** and **does not introduce extra computational overhead**. We additionally calculate the computation time for GRPO and our method in a training step.
> | Method                   | time  |
> |--------------------------|-------|
> | 1B+GRPO  |   9.54s  |
> | 1B+our |  9.54s + 0.15s  |
>
> We have added other models' time consumption of each training step in **Table 2** of the revised manuscript.
>
> > **W2:** The performance of the 1B model drops on the GenEval benchmark in Table 1. What is the reason?
>
> We may need to clarify this further. As shown in **Table 1**, the 1B model with our method outperforms the 1B model with GRPO in the **overall score and in six sub-tasks on Geneval**, with only the Color Attr. sub-task being slightly lower than that of GRPO. We argue that our method can be considered effective for the 1B model on the GenEval benchmark.
>
> > **W3:** Figure 7(a) only shows the effect of the selection ratio for each token type, while the analysis does not address the choice of the total selection proportion (e.g., 30%).
>
> (1) Our motivation is to identify critical tokens in RL-based AR visual generation. The core issue is to **clarify the types of critical tokens and validate their effectiveness**. We comprehensively identify which tokens are more critical for RL training from three perspectives, and provide extensive experiments of our method’s effectiveness across multiple models and benchmarks.
>
> (2) In the ablation experiments shown in **Figure 7**, we compare multiple ratios for each type of critical token. For each type, 10% serves as a performance gain threshold. Beyond this point, performance improves slowly. Therefore, we adopt 10% as the default setting for each type.
>
> (3) We provide experiments on the total selection proportion, including 15% (5%+5%+5%) and 45% (15%+15%+15%). We observe that reducing the selection proportion to 15% results in a significant performance drop, while **further increasing it does not bring notable gains**. These results correspond to the ablation experiment in Fig. 7. Thus, we adopt 30% (10%+10%+10%) as our default setting.
> | Method                       | DEQA↑ | HPS↑  | ImgReward↑ | PickScore↑ | GenEval↑ |
> |:-----------------------------|:-----:|:-----:|:----------:|:----------:|:--------:|
> | 15% (5%+5%+5%)               | 3.63  | 29.29 | 0.63       | 21.55      | 0.82     |
> | 30% (10%+10%+10%) (our)      | **3.73**  | 29.61 | 0.73       | 21.60      | **0.85**    |
> | 45% (15%+15%+15%)            | 3.70  | **29.65** | **0.75**       | **21.62**     | **0.85**     |
>
> We have added a more detailed analysis and discussion of the total selection proportion in **Appendix E.4** of the revised manuscript.
>
> > **Q1:** Since critical tokens may overlap across the defined categories, does this situation require additional processing?
>
> Thanks for your valuable question. In the implementation, we have already taken this into consideration. Specifically, we first select 10% of the initial tokens. Then, from the unselected tokens, we select 10% of the tokens with the highest entropy gradient. Finally, we select 10% of the remaining tokens with the lowest similarity.

---

> ### Author Response · Authors · 2025-11-28
> **Welcome for Further Feedback**
>
> Dear Reviewer z8AU,
>
> We sincerely thank you for your thorough and constructive feedback, which greatly guided our additional analyses during the rebuttal period. Following your insightful comments, we have conducted new experiments and ablation studies as presented in our **Rebuttal**. These updates have been incorporated into the revised manuscript and appendix to more clearly address each of your points.
>
> - **Regarding total selection proportion:** we provide experiments on the total selection proportion and observe that reducing the total selection proportion results in a significant performance drop, while **further increasing it does not bring notable gains**. These results are included in **Appendix E.4**.
>
> We hope that these substantial revisions fully address your thoughtful questions and concerns. If there are any remaining issues that require further clarification, we would be grateful for the opportunity to continue the discussion. Thank you again for your time, expertise, and constructive guidance.

---

### Official Review · Reviewer_MkTp · 2025-11-01

**Soundness:** 2
**Presentation:** 2
**Contribution:** 2
**Rating:** 4
**Confidence:** 2

**Summary:**

This paper proposes GCPO, which selects critical tokens through multiple criteria to improve GRPO for T2I generation. The authors identify the critical tokens as initial tokens, tokens with high entropy gradients, and tokens with low average similarities. A dynamic advantage weight is also proposed to balance exploration and stabilization.

**Strengths:**

1. The idea is intuitive and well-motivated.
2. Section 4 provides solid experimental observations, which support the proposed criteria for selecting critical tokens.
3. The proposed method is simple and appears to be effective in improving generation performance.

**Weaknesses:**

1. Figure 7 shows that “30% Critical Tokens with DAW” performs better than the “100% baseline,” while “30% Critical Tokens” performs worse. How does the model perform when DAW is applied to the “100% baseline”? It is important to determine whether DAW or the critical tokens contribute more to the overall performance.
2. To my understanding, $Z_{struct}$ is selected within a single image, while $Z_{sim}$ is selected across all images. Should $Z_{struct}$ therefore come from different positions for each image, and $Z_{sim}$ from the same positions across images? If so, why does $Z_{sim}$ come from different positions as shown in Figure 5?
3. The selection ratio (along with the ratio of each token type) appears to be an important hyperparameter. The current setting is 10%:10%:10%. It would be better if the authors could provide results for additional selection ratio settings.

**Questions:**

1. How does the group size G affect performance? Does it have a strong positive correlation?
2. Is the local averaging of the entropy map very important? It is intuitively reasonable, but it would be better to include an ablation study to verify its effect.

---

> ### Author Response · Authors · 2025-11-21
> **Response to Reviewer MkTp [1/2]**
>
> We sincerely thank the reviewer for the insightful comments. Before addressing your concerns, we would like to emphasize that our method, **for the first time**, reveals the three unique perspectives of critical tokens in the RL of autoregressive image generation, **providing valuable insights for further improving RL post-training for AR models.** This has been positively recognized by other reviewers.
>
> Our main experiments collectively validate the effectiveness of our method, and the additional experiments presented below further demonstrate the rationality of our design and the extension of our method (e.g., scale-wise model) in **Section 6.4**.
>
> And **Reviewer z8AU, Reviewer KyYg, and Reviewer eipe all give positive assessments of our work.** We sincerely hope our response below can address your concerns.
>
> > **W1:** How does the model perform when DAW is applied to the “100% baseline”? It is important to determine whether DAW or the critical tokens contribute more to the overall performance.
>
> We would like to provide further explanation to clarify the value of Dynamic Advantage Weight (DAW).
>
> (1) DAW is designed to work in cooperation with the critical token strategy and improve performance by further encouraging diverse advantages among the critical tokens. Applying DAW to all the 100% tokens may amplify certain negative gradients among the remaining non-critical tokens. To verify this, we conduct the corresponding experiments in **Table 8** and the following table.
>
> | Method                                 | DEQA↑ | HPS↑  | ImgReward↑ | PickScore↑ | GenEval↑ |
> |--------------------------------------|:-----:|:-----:|:----------:|:----------:|:--------:|
> | 100% Tokens without DAW (GRPO)         | 3.67  | 29.16 | **0.73**       | 21.59      | 0.84     |
> | 100% Tokens with DAW                   | 3.71  | 29.54 | 0.72       | 21.62      | **0.85**     |
> | 30% Critical Tokens without DAW        | 3.70  | 29.33 | 0.71       | 21.58      | 0.83     |
> | 30% Critical Tokens with DAW (our)     | **3.73**  | **29.61** | **0.73**       | **21.60**      | **0.85**     |
>
> (2) The experimental results show that our method **(30% tokens with DAW)** > (100% tokens with DAW) > (100% tokens without DAW). This verifies that **1)** DAW is also useful to the general GRPO. **2)** Critical tokens and DAW work synergistically to achieve the best performance. Therefore, the rational selection of critical tokens is essential for performance improvement, and combining this selection with DAW leads to even better results.
>
> (3) In addition, the comparison between (30% tokens without DAW) and (100% tokens with DAW) differs in both token numbers and DAW strategy, and thus is not a fair comparison. Thus, this is for reference only.
>
> (4) Notably, we want to emphasize that, even without DAW, the proposed critical token strategy is quite insightful and can match the performance of the full sequence GRPO.
>
> We have added a more detailed analysis and discussion of the DAW ablation in **Appendix E.1** of the revised manuscript.
>
> > **W2:**  Why does $Z_{sim}$ come from different positions as shown in Figure 5?
>
> Thanks for your careful review and for pointing this out. Your understanding is correct. We have revised our framework in **Figure 5**, where $Z_{sim}$ comes from the same position in all images within the same group.
>
> > **W3:**  The current setting is 10%:10%:10%. It would be better if the authors could provide results for additional selection ratio settings.
>
> (1) The ablation experiments, as shown in **Figure 7**, further illustrate this point. We compare multiple ratios for each type of critical token. For each type, 10% serves as a performance gain threshold. Beyond this point, performance improves slowly. Therefore, we adopt 10% as the default setting for each type.
>
> (2) We further provide experiments with other selection ratios, as shown in **Table 9**. We observe that when the total selection ratio is 30% and the selection ratio is set to 15%:10%:5%, the performance decreases significantly even though the total number of selected tokens remains unchanged. These results correspond to the ablation experiment in **Figure 7**.
>
> | Method                          | DEQA↑ | HPS↑  | ImgReward↑ | PickScore↑ | GenEval↑ |
> |--------------------------------------|:-----:|:-----:|:----------:|:----------:|:--------:|
> | 30% (15%:10%:5%)                 | 3.66  | 29.54 | 0.62       | 21.54      | 0.83     |
> | 30% (10%:10%:10%) (our)          | **3.73**  | **29.61** | **0.73**       | **21.60**      | **0.85**     |
>
> (3) Notably, our motivation is to **identify critical tokens in RL-based AR visual generation, which remain unexplored**. The core issue is to clarify the types of critical tokens and validate their effectiveness. We have added a more detailed analysis and discussion of the selection ratios in **Appendix E.2** of the revised manuscript.

---

> ### Author Response · Authors · 2025-11-21
> **Response to Reviewer MkTp [2/2]**
>
> > **Q1:**  How does the group size G affect performance? Does it have a strong positive correlation?
>
> Thank you for pointing out this direction worthy of further analysis. Previous methods [1] [2] generally employ a group size of 4 or 8 for GRPO, limited by GPU memory. We argue that for the GRPO algorithm itself, a larger group size is empirically verified to yield more stable policy variance and a more diverse exploration space. For example, Brorl [3] points out that substantially increasing the group size of the LLM can further improve the performance of models that are experiencing learning stagnation. However, there are not many ablation analyses of the impact of group size on different metrics in AR visual generation.
>
> To further validate this on AR visual generation, we provide more experiments of our method on different group sizes, as shown in **Table 5**. In the following table, we find that group size and HPS have a strong positive correlation. DEQA, PickScore, and ImageReward show significant improvements from 4 to 8, but the improvement is minimal when the group size is further increased (from 8 to 12). Conversely, the GenEval only shows an improvement when the group size increases (from 8 to 12).
>
> | Setting| GPU Memory↓ | Training Time↓ | DEQA↑ | HPS↑ | ImgReward↑ | PickScore↑ | GenEval↑ |
> |:--------:|:-------------:|:---------------:|:-------:|:------:|:------------:|:------------:|:----------:|
> | G=4    | 39G         | 4.5h          | 3.73  | 29.61| 0.73       | 21.60      | 0.85     |
> | G=8    | 79G         | 9.2h          | 3.80  | 30.39| 0.82       | 21.70      | 0.86     |
> | G=12   | 115G        | 11.4h         | **3.81**  | **30.63**| **0.85**       | **21.72**      | **0.88**     |
>
> We have added a more detailed analysis and discussion of the group size in **Section 6.5** of the revised manuscript.
>
> > **Q2:**  Is the local averaging of the entropy map very important?
>
> Local averaging of the entropy map is proposed to reduce the noise and outliers in the entropy map, enabling more robust and efficient critical token selection. We further provide corresponding ablation experiments on this strategy in **Table 10** and the following table, where removing the local averaging results in a slight decrease in model performance.
>
> | Method                   | DEQA↑ | HPS↑  | ImgReward↑ | PickScore↑ | GenEval↑ |
> |--------------------------|:-----:|:----:|:----------:|:----------:|:--------:|
> | w/o local averaging      | 3.71 | 29.56 | 0.71       | 21.58      | 0.84     |
> | w local averaging (our)  | **3.73**  | **29.61** | **0.73**       | **21.60**      | **0.85**     |
>
> We have added a more detailed analysis and discussion of the local averaging of the entropy map in **Appendix E.3** of the revised manuscript.
>
> > **Reference**
>
> [1] T2I-R1: Reinforcing Image Generation with Collaborative Semantic-level and Token-level CoT
>
> [2] ReasonGen-R1: CoT for Autoregressive Image Generation model through SFT and RL
>
> [3] BroRL: Scaling Reinforcement Learning via Broadened Exploration

---

> ### Author Response · Authors · 2025-11-28
> **Welcome for Further Feedback**
>
> Dear Reviewer MkTp,
>
> We sincerely thank you for your thorough and constructive feedback, which greatly guided our additional analyses during the rebuttal period. Following your insightful comments on **W1-3** and **Q1–Q2**, we have conducted a substantial number of new experiments and ablation studies as presented in our **Rebuttal**. These updates have been incorporated into the revised manuscript and appendix to more clearly address each of your points:
>
> - **Regarding W1 (Dynamic Advantage Weight, DAW):** we have added experimental results on DAW. These analyses clearly demonstrate that DAW is beneficial not only for general GRPO but also for critical tokens, and that DAW and critical tokens work synergistically to achieve the best performance. These results are presented in **Table 8 and Appendix E.1**.
> - **Regarding W2:** we have revised our framework in **Figure 5,** where $Z_{sim}$comes from the same position in all images within the same group.
> - **Regarding W3 (additional selection ratio settings):** we further provide experiments with other selection ratios. We discuss that the performance under other **selection ratio settings** decreases significantly even though the total number of selected tokens remains unchanged. These results are included in **Table 9 and Appendix E.2**.
> - **Regarding Q1 (Group size):** we provide more experiments of our method on different group sizes and discuss their influence on various metrics for AR generation. These results are included in **Table 5 and** **Section 6.5**.
> - **Regarding Q2 (local averaging of the entropy map):** we provide corresponding ablation experiments to further demonstrate the rationale behind local averaging. These results are included in **Table 10 and Appendix E.3**.
>
> We hope that these substantial revisions fully address your thoughtful questions and concerns. If there are any remaining issues that require further clarification, we would be grateful for the opportunity to continue the discussion. Thank you again for your time, expertise, and constructive guidance.

---

### Author Response · Authors · 2025-11-29
**Official Comment by Authors About the Review**

Dear AC,

We sincerely appreciate the time and effort you have dedicated to reviewing our paper. To help you grasp the current status of the review, we would like to provide a summary of the reviewers' comments.

We are encouraged that **Three Reviewers z8AU, KyYg, and eipe** have provided **positive ratings with high confidence**. Additionally, **Reviewer MkTp** assigned a score with **low confidence**, primarily due to considerations regarding supplementary experiments to further enhance the comprehensiveness of our method.

We appreciate that the reviewers consistently acknowledge our work, specifically highlighting the following strengths:

- The proposed method is **well-motivated** and offers **valuable insights** to improve RL post-training for AR models.
- The proposed method demonstrates **significant performance improvements**, supported by convincing experiments across **multiple AR models and T2I benchmarks**.

During the rebuttal phase, we addressed the constructive comments from the reviewers:

- **Additional experimental settings:** We demonstrated that our default setting is both reasonable and effective compared to more configurations, such as other parameters.
- **More generation paradigms:** We showed that our method is also effective for other generative paradigms, such as scale-wise model (VAR-based).
- **Time consumption comparison:** We confirmed that our method introduces no additional computational overhead.

We once again thank the reviewers and the AC for your time and dedication. We believe that our efforts during the rebuttal, guided by these constructive suggestions, will effectively address the reviewers' concerns, assist the AC in assessing the value of our work, and allow us to enhance the comprehensiveness and rigor of the paper.

Best regards,

Authors of Submissions 425

---

### Meta-Review · Area_Chair_NR9K · 2026-01-05

**Summary:**

This paper proposes GCPO to focus on critical tokens during reinforcement learning for image generation. The method is simple, well-motivated, and achieves good results across various benchmarks. Some reviewers worried about hyperparameter sensitivity and the specific impact of the weighting strategy. There were also concerns about small gains on some tasks and missing model comparisons. Initial reviews were split regarding the method's novelty and efficiency. The rebuttal addressed these with new experiments showing the approach is fast and effective. Overall this is a solid paper and the authors are encouraged to explore more diverse model architectures further.

**Reviewer Concerns:**

The rebuttal addressed concerns about training time and the interaction between tokens and weighting. The authors also clarified token selection and added tests for multi-scale models. However, some reviewers still felt the performance gains were small on certain tasks. The theoretical link between entropy and image structure remains partially unexplored.

**Reviewer Scores:**

Reviewer MkTp is less confident and would likely have raised their score after seeing the new ablation results. Reviewer z8AU would have remained positive given the low computational overhead. Reviewer KyYg would have kept a positive rating due to the successful scaling tests. Reviewer eipe would have stayed at a marginal accept after the authors included discussion on alternative paradigms.

---

### Decision · Program_Chairs · 2026-01-26

Accept (Poster)